# Flame-Retardant and Smoke-Suppression Properties of Bamboo Scrimber Coated with Hexagonal Boron Nitride

Gaihuan Li [1], Ying Yu [1], Shaofei Yuan [2], Wenfu Zhang [2] and Xinzhou Wang [1,3,*]

1   Co-Innovation Center of Efficient Processing and Utilization of Forest Resources, College of Materials Science and Engineering, Nanjing Forestry University, Nanjing 210037, China; 15933071663@163.com (G.L.); 15380420802@163.com (Y.Y.)
2   Zhejiang Academy of Forestry, Hangzhou 310023, China; fei20008281@126.com (S.Y.); zhangwenfu542697@163.com (W.Z.)
3   Fujian Jingmingsheng Forestry Technology Co., Ltd., Sanming 366035, China
*   Correspondence: xzwang@njfu.edu.cn

**Abstract:** In order to improve the flame-retardant properties of bamboo scrimber, *chitosan* (CS) and *polyvinyl alcohol* (PVA) were used as the film-forming substances, and *hexagonal boron nitride* (h-BN) was used as the flame-retardant substance to prepare h-BN flame-retardant coatings, which were coated on the surface of the bamboo scrimber. The effects of the h-BN flame-retardant coatings with different quality concentrations on the flame-retardant properties of the bamboo scrimber, as well as on the morphology of the residual carbon, were investigated using the analytical methods of FTIR, environmental scanning electron microscopy, thermogravimetric analysis, combustion test, and coating adhesion test. The results showed that the h-BN flame-retardant coating could improve the thermal stability of the bamboo scrimber and that the higher the mass concentration, the better the thermal stability of the h-BN. Compared to the control, the time to ignition (TTI) of the 5% h-BN flame-retardant-treated specimens increased by 56%; the peak heat release rate (Pk-HRR), total heat release (THR), and total smoke production (TSP) decreased by 9.92%, 7.54%, and 32.35%, respectively; however, due to the presence of PVA, the peak smoke production rate (Pk-SPR) increased by 17.78%. The 5% h-BN coating had very good adhesion, with an adhesion grade of zero. In conclusion, the h-BN coating could be well-adhered to the surface of the bamboo scrimber, and the 5% h-BN flame-retardant coating had a better flame retardancy compared to other treatments, meaning that it could provide a new strategy for improving the flame-retardant properties of bamboo scrimber for construction use.

**Keywords:** bamboo scrimber; hexagonal boron nitride; thermal stability; flame retardancy

## 1. Introduction

Bamboo scrimber is a bamboo-based fiber composite material made of bamboo bundles as the basic component, glued together in a parallel arrangement [1]. As a novel material with the advantage of green environmental protection, bamboo scrimber has excellent mechanical properties and is widely utilized in interior decoration and architecture [2,3]. As the main processing raw material of bamboo scrimber is bamboo, which is combustible, bamboo scrimber also has the characteristic of being easy to burn [4], and this defect makes the application of bamboo scrimber in interior decoration and architecture subject to certain limitations. To resolve this problem, it is necessary to improve the flame retardancy of bamboo scrimber using a flame-retardant treatment.

Currently, the research conducted by scholars in various countries on bamboo scrimber is mostly about its physical properties [5], mechanical properties [6–8], mildew resistance [9,10], surface properties [11], etc., and there are relatively few researches on its flame-retardant properties. Most flame-retardant treatments are impregnation treatments utilizing phosphorus–nitrogen flame retardants or nitrogen–phosphorus–boron composite

flame retardants: e.g., Ran et al. [12] impregnated the bamboo scrimber with different concentrations of flame retardant CaAl-PO4-LDHs. The combined results demonstrated that the synthesized CaAl-PO4-LDHs significantly improved the flame retardancy of bamboo scrimber. Du et al. [13] explored the effect of the low-concentration ammonium polyphosphate impregnation (APP) sequence on the combustion performance of bamboo scrimber. These results revealed that the thermal stability of bamboo bundles was significantly increased after being immersed in APP.

Although this treatment plays a flame-retardant role, there are problems such as the large amount of flame retardant required, the high cost, the poor environmental protection, and the impact on the physical and mechanical properties of the material [14]. In contrast, as one of the well-established and most efficient methods, the surface coating method has a simple operation [15], a low flame retardant dosage, the ability to maintain the overall performance of the bamboo scrimber materials, and the applicability to a wide range of substrates, such as wood [16], textiles [17], and polymers [18].

The literature has demonstrated that *hexagonal boron nitride* (h-BN) is a good flame-retardant substance that can effectively improve the flame-retardant properties of materials [19]. h-BN coatings have good anisotropy [19], which can provide rapid in-plane heat diffusion at high temperatures or in combustion environments [20], slowing down heat conduction through the dense material, thus improving the material's ignition performance and effectively prolonging the material's ignition time [21]. However, most flame-retardant coatings have poor adhesion.

*Chitosan* (CS) is a natural polysaccharide cellulose, widely found in the shells of insects and crustaceans and in the cell walls of fungi, also known as soluble chitin, which has become highly researched in recent years owing to its renewability, non-toxicity, and high biocompatibility [22], but the stronger intramolecular and intermolecular hydrogen bonding and the low crystallinity lead to the brittleness of CS membranes [23,24]. As a novel green material in recent years, *polyvinyl alcohol* (PVA) has good film-forming properties, chemical stability, solvent resistance, and strong adhesion [25]. Therefore, PVA and CS are commonly blended and adopted as film-forming substances [26,27] which not only have good film-forming properties but also possess a strong bonding ability.

Here, PVA and CS were utilized as film-forming substances, and h-BN flame-retardant coatings were prepared by adding different mass concentrations of h-BN. The structure and surface morphology of the coatings were tested using FTIR and environmental scanning electron microscopy. The pyrolysis behavior and combustion performance tests were conducted using a thermogravimetric analyzer and cone calorimetry, providing a basis for fabricating bamboo scrimber with flame-retardant properties.

## 2. Materials and Methods

### 2.1. Materials

(1) Bamboo scrimber: It was purchased from Fujian Jingmingsheng Forestry Technology Co. (Sanming, China). The air-dry density was 1.03 g/cm$^3$ and the moisture content was 2.3%. The bamboo scrimber air-dry and moisture content determination standards reference the national standard GB/T 1931-2009 [GB/T 1931-2009 Method for determination of the moisture content of wood. TC41. 2009, China] wood moisture content determination method. The bamboo was processed into a number of $100 \times 100 \times 13$ mm specifications of the sheet.

(2) Raw materials for the flame-retardant coating treatment: The test materials required for the preparation of the flame-retardant coating included hexagonal boron nitride, acetic acid, chitosan, and polyvinyl alcohol, and the detailed parameters of the test materials are shown in Table 1.

**Table 1.** Raw materials for the coating treatment.

| Title | Model/Grade | Production Unit |
|---|---|---|
| Acetic acid ($CH_3COOH$) | Analytical purity | Sinopharm Chemical Reagent Co. (Shanghai, China) |
| Chitosan | Deacetylation degree $\geq$95 | Shanghai McLean Biochemical Technology Co. (Shanghai, China) |
| Polyvinyl alcohol | 1788 | Shanghai McLean Biochemical Technology Co. (Shanghai, China) |
| Hexagonal boron nitride | Particle size 25 $\mu$m | Suzhou Yuante New Material Co. (Suzhou, China) |

### 2.2. Preparation of the h-BN Flame-Retardant Coating

First, 92 g of aqueous acetic acid solution with a concentration of 3% was prepared, and then 4 g of CS and 4 g of PVA were added and stirred utilizing a thermostatically heated magnetic stirrer with the temperature condition set at 98 °C and stirred at 500 rpm until the CS and PVA were completely dissolved. The CS/PVA solution was left to stand and then allowed to cool down to room temperature, and 100 mL of a h-BN aqueous solution with different mass concentrations (1%, 5%, 10%) of h-BN was added, stirred at 1500 rpm for 30 min utilizing a magnetic stirrer and left to stand.

The coat-spraying method was chosen to spray the coat on the surface of the bamboo scrimber utilizing an electric spray gun. The amount of coat per plate was 10% of the weight of the plate, and it was left to dry naturally after treatment. The dry film thickness was 0.3 mm $\pm$ 0.02 mm.

### 2.3. Analysis and Testing

#### 2.3.1. Fourier Transform Infrared Spectroscopy (FTIR) Test

The samples were analyzed for functional groups on the surface utilizing an infrared spectrometer (BRUKER, USA, model VERTEX80V) with a scanning range of 4000–400 cm$^{-1}$ and 15 scans at a resolution of 4 cm$^{-1}$. The attenuated total reflection (ATR) test mode was utilized to determine pure coatings. The samples were prepared by uniformly pouring the same mass of pure coatings onto glass Petri dishes, after which they were dried. After that, the dried flakes were cut with scissors to a size of 5 $\times$ 5 mm.

#### 2.3.2. Environmental Scanning Electron Microscope-X-ray Energy Spectrum Test

The surface and internal morphology of the bamboo scrimber treated with different mass concentrations of h-BN flame retardant, and the surface and internal morphology of the charcoal residue after combustion were observed using an environmental scanning electron microscope (Thermo Fisher Scientific, Shanghai, China, model Quanta 200) equipped with an energy dispersive X-ray analyzer (EDS). Before the experiment, samples of 10 $\times$ 10 $\times$ 10 mm were prepared and gold-sprayed.

#### 2.3.3. Thermal Stability Test

The thermal stability of the samples was tested utilizing a thermogravimetric analyzer (NETZSCH Scientific Instruments, German, model TG 209 F3). The mass of the samples was in the range of 5–10 mg. The temperature range was between 25 °C and 800 °C, with a heating rate of 10 °C/min. The test was conducted under a dry nitrogen environment with a flow rate of 20 mL/min.

For the test, we labeled the untreated bamboo scrimber as URB, theBN flame-retardant-treated bamboo scrimber as FRRB$_1$, the 5% h-BN flame-retardant-treated bamboo scrimber as FRRB$_2$, and the 10% h-BN flame-retardant-treated bamboo scrimber as FRRB$_3$, which they will be referred to as hereinafter.

#### 2.3.4. Flame-Retardant Property Test

The samples were placed at a temperature of (23 $\pm$ 2) °C and a relative humidity of (50 $\pm$ 5)% until the mass was constant, and then the samples were tested for flame retardancy.

The combustion performance of the samples was measured utilizing a cone calorimeter (FTT Company, UK, model FTT2000), and the test was conducted in the spirit of the ISO5660-1:2015 [ISO5660-1:2015 Heat release rate (cone calorimeter method) and smoke

production rate (dynamic measurement). ISO/TC 92/SC 1. 2015, UK] combustion reaction test standard, with the setting of thermal radiation intensity to 50 kW/m$^2$, a temperature of 740 °C, a time of 600 s, and a sample size of $100 \times 100 \times 13$ mm. The time to ignition (TTI), heat release rate (HRR), total heat release rate (THR), smoke release rate (SPR), total smoke volume, COP, and $CO_2P$ were obtained from the test to analyze the combustion performance of the coating.

A pre-testing of the flame-retardant test was conducted and found to be stable, probably because of the good homogeneity and compactness of the bamboo scrimber [28] and the precise control during the preparation of the coating to ensure the consistency of the treatments. Therefore, the test was conducted on only one sample per series, as is sometimes the case in flame-retardant tests [14,29]. Thus, the findings of the flame-retardant study in this experiment could be considered indicative to conduct further tests in the future to understand the details of the experiments and corroborate the data.

### 2.3.5. Coating Adhesion Test

The coating adhesion test was conducted according to the GB/T 4893.4-2013 [GB/T 4893.4-2013 Test of surface coatings of furniture—Part 4: Determination of adhesion—Cross cut. SAC/TC 180. 2013, China] standard methods by sticking the 3M transparent tape firmly and then tearing it off quickly at an angle of 60°, utilizing a magnifying glass to observe the coating peeling off and judging its grade. There were 0–5 adhesion test levels, where Level 0 represented the best adhesion.

## 3. Results

### 3.1. Analysis of Chemical Structure of Flame-Retardant Coatings

Figure 1 illustrates the FTIR profiles of the h-BN coatings with different mass concentrations. The FTIR structural analysis of the flame-retardant coatings revealed that the FTIR curves of the h-BN flame-retardant coatings with mass concentrations of 1%, 5%, and 10% were adopted with absorption peaks of CS, PVA, and h-BN. For example, the characteristic peak at 2921 cm$^{-1}$ was generated by the presence of methyl groups (-CH3) and alkyl side chains (-CH2-) on the sugar ring of CS. The strong, broad peak at 3261 cm$^{-1}$ may be related to the -NH2 group of CS or the presence of -OH groups in the PVA and CS. The B-N in-plane telescopic vibration peak and the B-N-B out-plane bending vibration peak of h-BN were 1338 cm$^{-1}$ and 771 cm$^{-1}$, respectively. This initially indicated that the h-BN flame-retardant coatings were successfully prepared [30].

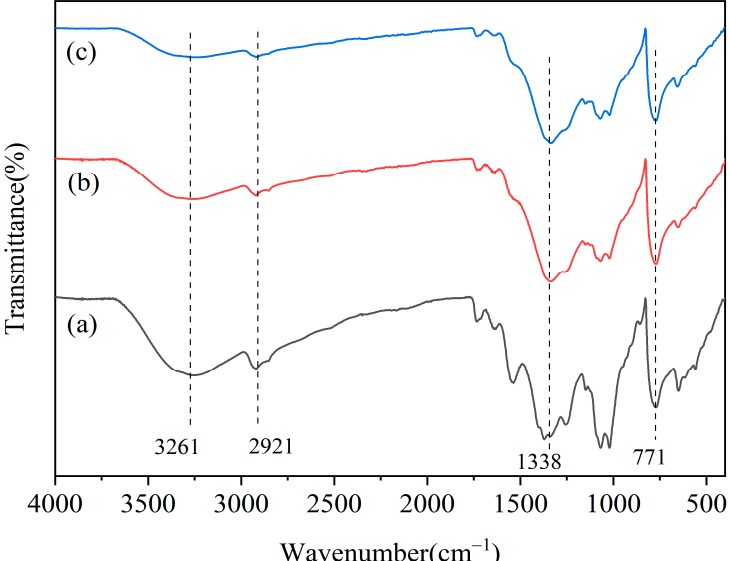

**Figure 1.** FTIR spectra of h-BN coatings with different mass concentrations. Note: (**a**) 1% h-BN; (**b**) 5% h-BN; (**c**) 10% h-BN.

### 3.2. Morphology and Elemental Analysis of Flame-Retardant Coatings on Bamboo Scrimber Surfaces

Figure 2 reflects the surface morphology and distribution of the B element in bamboo scrimber treated with different mass concentrations of h-BN. From the SEM images, the surface of the bamboo scrimber treated with the flame-retardant treatment did not demonstrate the typical surface characteristics of bamboo scrimber, which proved that the flame-retardant coating completely covered the surface of the samples. The flame-retardant coating had a dense and smooth surface. The comparison of the SEM image (a1)–(d1) reveals that, when the mass concentration of h-BN was 10%, the coating surface demonstrated a significant agglomeration phenomenon. The distribution figures of the B elements (a2)–(d2) reveal that the B element contents on the surface of the untreated, 1%, 5%, and 10% h-BN flame-retardant-treated bamboo scrimber were 0%, 16.8%, 18%, and 17.4%, respectively. The B element content was highest at 5% instead of 10%, which may be because, at 10%, h-BN was encapsulated by film-forming material, which led to the appearance of an agglomeration phenomenon, leading to the difference between the detected content and the actual content on the surface. Figure 2(b2),(c2) depict that the dispersion of h-BN in the film-forming material was more uniform when the mass concentration of h-BN was 1% and 5%, while the dispersion of the B element in the film-forming material was lower when the mass concentration of h-BN was 10%, as illustrated in Figure 2(d2). This indicated that the higher the mass concentration of h-BN, the more serious the aggregation phenomenon between the particles, resulting in the uneven distribution of h-BN on the coating surface.

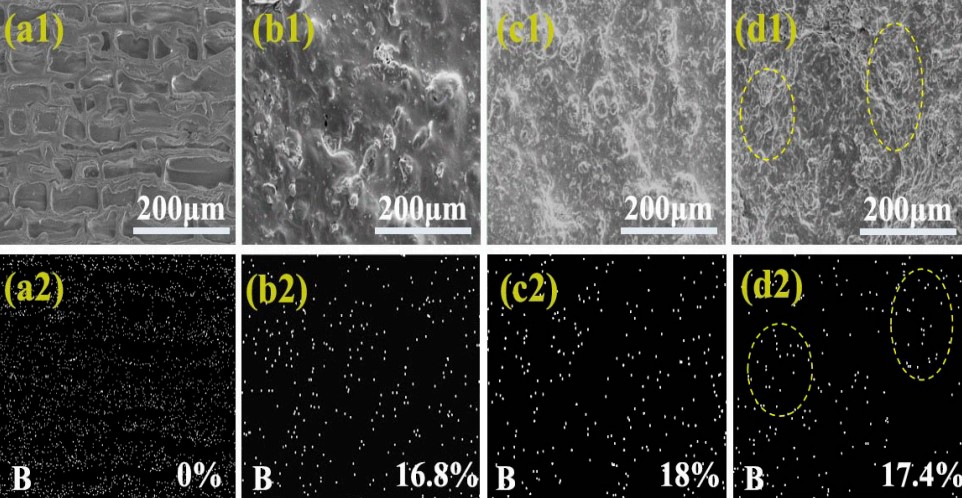

**Figure 2.** SEM images and Element B energy spectra of bamboo scrimber surface coated with different mass concentrations of h-BN. (**a1,a2**), (**b1,b2**), (**c1,c2**), and (**d1,d2**) are SEM images and Element B energy spectra of bamboo scrimber surface coated with untreated, 1%, 5%, and 10% h-BN, respectively.

### 3.3. Thermal Stability Analysis

The TGA test was conducted to determine the thermal performance of the prepared flame-retardant coatings and flame-retardant bamboo scrimber samples. Figure 3a reflects the heat resistance of the flame-retardant coatings with different mass concentrations of h-BN. There was only a clear platform of the degradation behavior of the synthesized coatings from the thermogravimetric graphs. At the end of the thermogravimetric test, the mass residuals of the pure coatings with 1%, 5%, and 10% h-BN mass concentration were 39.21%, 50.86%, and 61.30%, in that order, indicating that the increase in h-BN concentration enhanced the thermal stability of the coatings. The reason for this was that h-BN has an excellent unidirectional thermal conductivity [31] and was dispersed into a layered structure in the film-forming material [32]. It allowed the thermal conductivity

path of the coating to be extended, the heat transfer to be effectively impeded, and the thermo-mechanical properties to be improved.

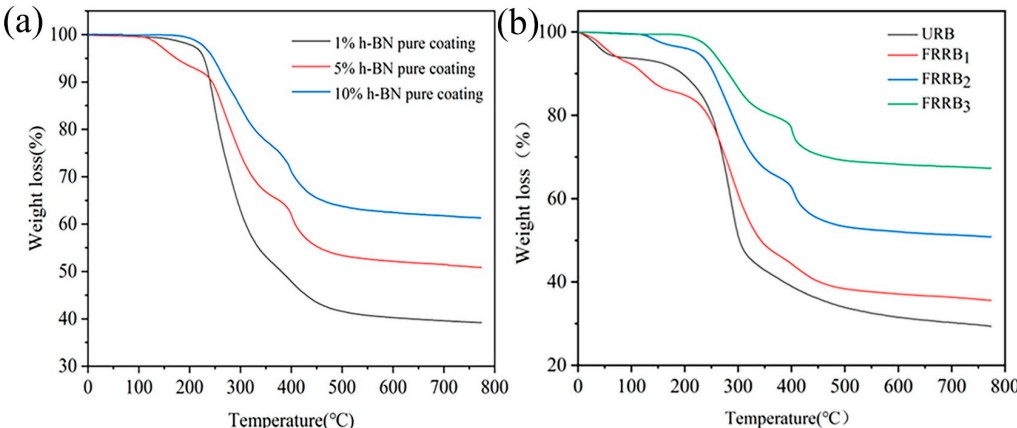

**Figure 3.** TGA curves of the h-BN pure coatings and flame-retardant bamboo scrimber samples. (**a**) TGA curves of the pure coating samples; (**b**) TGA curves of the flame-retardant bamboo scrimber samples. Note: URB—untreated bamboo scrimber; $FRRB_1$—1% h-BN flame-retardant-treated bamboo scrimber; $FRRB_2$—5% h-BN flame-retardant-treated bamboo scrimber; $FRRB_3$—10% h-BN flame-retardant-treated bamboo scrimber; and the same below.

Figure 3b is the TGA curve of the flame-retardant bamboo scrimber coated with different mass concentrations of h-BN, reflecting the effects of different mass concentrations of h-BN on the thermal stability of the bamboo scrimber samples. The slope of the mass loss rate curve of the bamboo scrimber gradually decreased with increasing h-BN concentration, the thermal degradation slowed down, and the catalytic charcoal formation of the coating was enhanced [32]. At the end of the thermogravimetric test, the mass residual rate of the untreated material was 29.4%, and the mass residual rate of the bamboo scrimber coated with different mass concentrations (1%, 5%, and 10%) of h-BN was 35.60%, 50.83%, and 67.30%, in that order. The presence of the h-BN coating improved the thermal stability of the bamboo scrimber. The higher the mass concentration of h-BN, the greater the effect on the thermal stability of the bamboo scrimber, because h-BN has good fire resistance and, therefore, was not easy to burn. The h-BN was dispersed as a laminated substance in the film-forming substance, which prolonged the path of heat conduction and increased heat consumption in the film-forming substance CS/PVA. Thus, the test result indicated that adding h-BN could hinder the heat propagation and slow down the rate of degradation of the bamboo scrimber substrate.

### 3.4. Flame-Retardant Performance Analysis

#### 3.4.1. Combustion Performance

The combustion performance of the bamboo scrimber coated with different quality concentrations of h-BN coating was conducted. The time to ignition (TTI) is shown in Figure 4. The HRR curves, THR curves, SPR curves, TSP curves, COP curves, and $CO_2P$ curves are shown in Figure 5.

TTI refers to the sustained ignition time required to achieve flaming combustion on the surface of the material, and the longer the TTI, the less likely the material will ignite under the experimental conditions and the better the flame-retardant effect of the material. Compared to the untreated material, the ignition time of the 1% h-BN, 5% h-BN, and 10% h-BN flame-retardant-treated bamboo scrimber was improved to different degrees, by 17%, 56%, and 58%, respectively. It demonstrated that the 5% and 10% h-BN flame-retardant coatings could effectively prolong the ignition time and have a good flame-retardant effect. There was little difference in the flame-retardant effect between the 5% and 10% h-BN flame-retardant coatings.

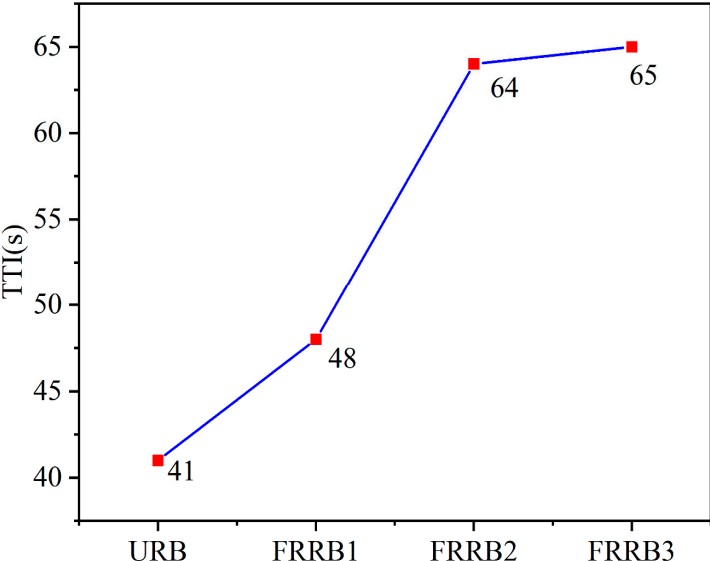

**Figure 4.** The time to ignition of the h-BN flame-retardant-treated bamboo scrimber.

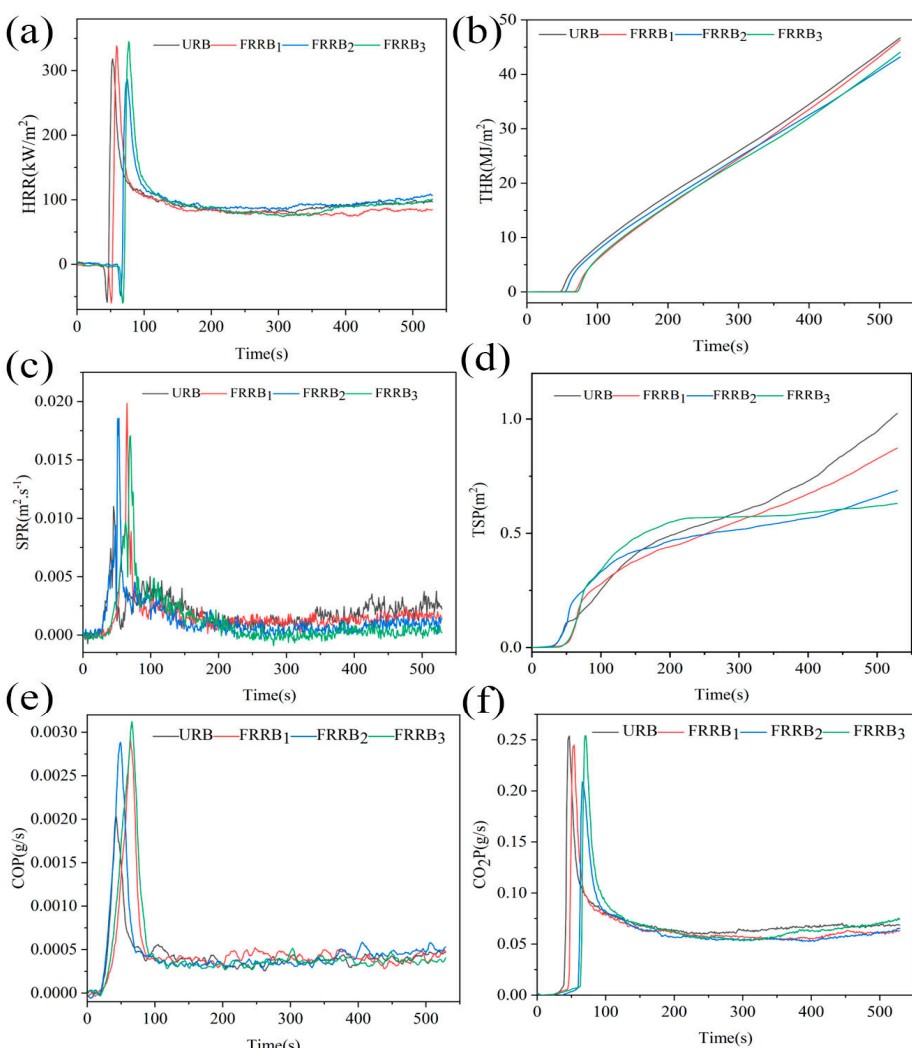

**Figure 5.** The cone calorimetric test results of the h-BN flame-retardant-treated bamboo scrimber cones: (**a**) HRR curves; (**b**) THR curves; (**c**) SPR curves; (**d**) TSP curves; (**e**) COP curves; and (**f**) $CO_2P$ curves.

HRR is the rate at which heat is released from a burning specimen per unit surface area and is one of the most important parameters for determining the fire risk of a material [29]. The greater the HRR, the greater the risk of a fire occurring in the material. Figure 5a illustrates that the maximum heat release rate of the 1% h-BN, 5% h-BN, and 10% h-BN flame-retardant-treated bamboo scrimber were 338.42 kW·m$^{-2}$·s$^{-1}$, 286.63 kW·m$^{-2}$·s$^{-1}$, and 345.03 kW·m$^{-2}$·s$^{-1}$, respectively, which decreased by 9.92% for the 5% h-BN flame-retardant-treated bamboo scrimber compared to the untreated material. The 1% and 10% h-BN flame-retardant-treated bamboo scrimber increased by 6.36% and 8.44%, respectively. The occurrence time of the maximum heat release rate of the 1% h-BN, 5% h-BN, and 10% h-BN flame-retardant-treated bamboo scrimber were prolonged by 11.32%, 41.51%, and 45.28%, respectively, compared to the untreated materials. This may be because the PVA and CS were dehydrated into charcoal during the combustion process, which effectively isolated the oxygen and heat transfer, and the addition of hexagonal boron nitride prolonged the heat conduction path, resulting in a delay in the emergence of the maximum heat release rate, such that the bamboo scrimber's combustion could be promptly dissipated before the emergence of the maximum heat release rate, slowing down the thermal degradation and decreasing the rate of combustion. From the perspective of the heat release rate, the most suitable concentration of the h-BN flame-retardant coating was 5%.

THR is the total heat released per unit area of a sample from the beginning to the end of combustion [33]. Combining the THR with the HRR allows for a more comprehensive evaluation of the combustion characteristics of materials [34]. As demonstrated in Figure 5b, the THR of the samples increased with time throughout the combustion process. Compared to the untreated material, the THR values of the 1% h-BN, 5% h-BN, and 10% h-BN flame-retardant-treated bamboo scrimber decreased by 0.86%, 7.54%, and 5.69%, respectively, which indicated that the addition of h-BN could reduce the total released heat of the bamboo scrimber and reduce the risk of fire. From the point of view of the total released heat, the most suitable concentration of the h-BN flame-retardant coating was 5%.

SPR refers to the amount of smoke a burning material releases during a fire [35]. It is commonly utilized to assess the extent of smoke production in a fire. A high SPR implies that a fire has produced a large amount of toxic smoke, which may lead to asphyxiation, intoxication, or obstruction of vision, thus interfering with escape and rescue operations. As illustrated in Figure 5c, compared to the untreated material, the peak smoke release rate of the 1% h-BN, 5% h-BN, and 10% h-BN flame-retardant-treated bamboo scrimber was delayed by 81.82%, 72.73%, and 54.55%, respectively, whereas, the peak smoke production rate was improved by 44.44%, 17.78%, and 55.56%, respectively. The smoke release was mainly caused by the PVA in the film-forming material burning when met with an open fire and releasing a large amount of smoke [25], but the h-BN could delay the smoke release of the bamboo scrimber.

TSP refers to the total amount of smoke produced during the burning process of the sample. As illustrated in Figure 5d, the highest TSP value for the untreated material was 1.02 m$^2$, and the TSP values for the bamboo scrimber with the 1% h-BN, 5% h-BN, and 10% h-BN flame-retardant treatments were reduced by 14.71%, 32.35%, and 38.24%, respectively, compared to the untreated material. With the increase in h-BN concentration, the total smoke release gradually decreased. This indicated that h-BN could effectively inhibit the smoke release during combustion and reduce the smoke release yield in the bamboo scrimber combustion process. Combining the smoke release rate and total smoke production indicators, the optimum concentration for flame retardancy was 5%; when the total smoke release was lower, the peak of smoke production rate was lowest, which could reduce the hazardousness caused by a fire.

The carbon monoxide and carbon dioxide release rate curves of the untreated and 1%, 5%, and 10% h-BN flame-retardant-treated bamboo scrimber during combustion are illustrated in Figure 5e,f, respectively. Figure 5e demonstrates that the CO release rate of the h-BN flame-retardant-treated bamboo scrimber was increased compared to that of the

untreated material. The h-BN flame-retardant coating could not inhibit the smoke's release rate completely and efficiently. The reason may be related to the incomplete oxidation of the volatile substances decomposed by h-BN in the bamboo scrimber, which could not come into contact with enough $O_2$, increasing the amount of CO. Figure 5f demonstrates that the $CO_2$ concentration change curve and the HRR curve are similar, and that, after the h-BN flame-retardant treatment, the $CO_2$ release rate of the bamboo scrimber's combustion is reduced, with the 5% h-BN flame-retardant treatment showing the most obvious reduction in the rate of $CO_2$ released. Hence, the most suitable concentration of the h-BN flame-retardant coating was 5%.

### 3.4.2. Morphology of Samples after Flame Retardancy Tests

To further investigate the flame-retardant mechanism of the h-BN flame-retardant coating on bamboo scrimber, morphological and structural analyses were conducted on the surface and interior of the carbon layer of the bamboo scrimber tested using the environmental scanning electron microscope, and the results are presented in Figure 6. As illustrated in Figure 6(a1)–(d1), the carbon layer of the untreated bamboo scrimber had a loose structure, which was difficult to keep in the structure of the bamboo scrimber, whereas, for the bamboo scrimber after the flame-retardant treatment with h-BN, the structure of the carbon layer was relatively tighter and more complete, indicating that the coating could increase the support time of the sample during the burning process. Simultaneously, a white substance precipitation was clearly observed on the surface of the fire-retardant-treated carbon layer. The surface components of the carbon layer were PVA, CS, and h-BN, but the component which had a white color was mainly h-BN; there may also have been a tiny amount of PVA. As a good heat-resistant and super-hard material [32], the layered h-BN played a role in enhancing the stability of the carbon layer during combustion, and, simultaneously, when burning, it migrated and covered the surface of the material, which prevented the material from further pyrolysis and combustion, and played a certain role in flame retardancy and smoke-suppression.

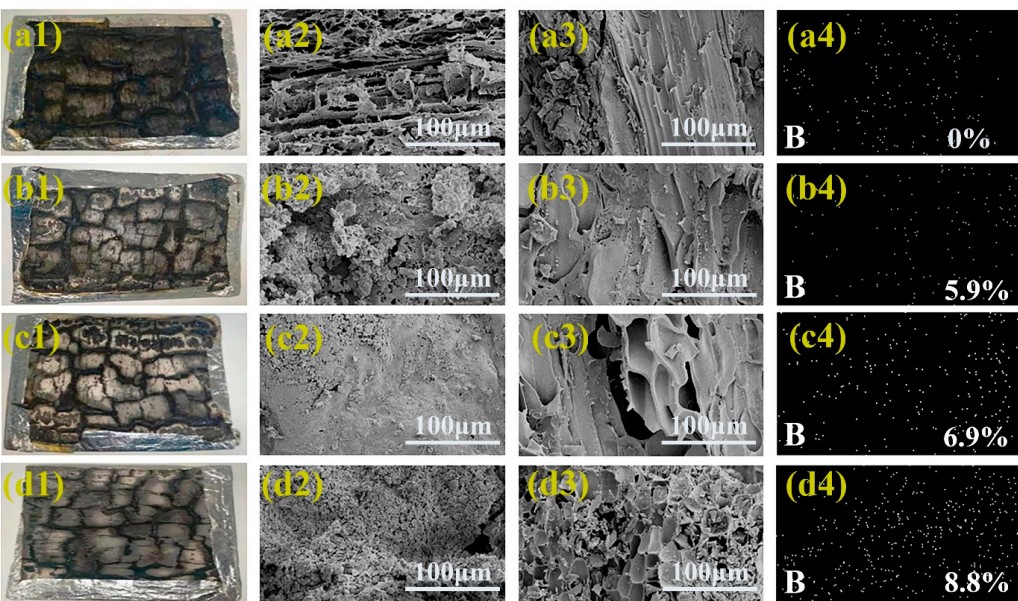

**Figure 6.** Digital photographs and SEM images of the carbon residues of the coatings after the cone calorimeter test. Note: (**a1**)–(**a4**): untreated bamboo scrimber; (**b1**)–(**b4**): 1% h-BN flame-retardant-treated bamboo scrimber; (**c1**)–(**c4**): 5% h-BN flame-retardant-treated bamboo scrimber; and (**d1**)–(**d4**): 10% h-BN flame-retardant-treated bamboo scrimber.

Figure 6(a2)–(d2) represent the surface morphology of the untreated, 1%, 5%, and 10% flame-retardant-treated bamboo scrimber carbon residues. Figure 6(a3)–(d3) represent the

internal morphology of the untreated, 1%, 5%, and 10% flame-retarded-treated bamboo scrimber carbon residues. Figure 6(a2)–(d2) and Figure 6(a3)–(d3) depict that the untreated bamboo scrimber burnt thoroughly. As for the bamboo scrimber treated with the h-BN flame-retardant treatment, the material's surface was smoother after combustion, and the carbon layer was retained intact and solid. Among them, the 5% flame-retardant-treated carbon layer had the best surface smoothness, which corroborated that the 5% h-BN was most uniformly dispersed in the film-forming material and that the presence of h-BN could act as a barrier between the flame and the substrate material. This carbon layer could better protect the substrate material [32].

According to the results of the B-element mapping in Figure 6(a4)–(d4), the content of the B-element in the carbon layer of the untreated material was 0%, while the surface of the carbon layer treated with the 10% h-BN flame-retardant had the highest content of B-element, which accounted for 8.8%, followed by the 5% flame-retardant-treated layer, which accounted for 6.9%, and the least B-element content was found in the 1% flame-retardant-treated layer, which accounted for 5.9%. This indicated that the component that played the role of flame retardant was h-BN.

### 3.5. Coating Adhesion

In practical applications, there are certain requirements for the adhesion of the coating; thus, it is necessary to characterize and test the adhesion of flame-retardant coatings, and the test results are presented in Table 2. The adhesion tests of the coatings applied to the bamboo scrimber are illustrated in Figure 7.

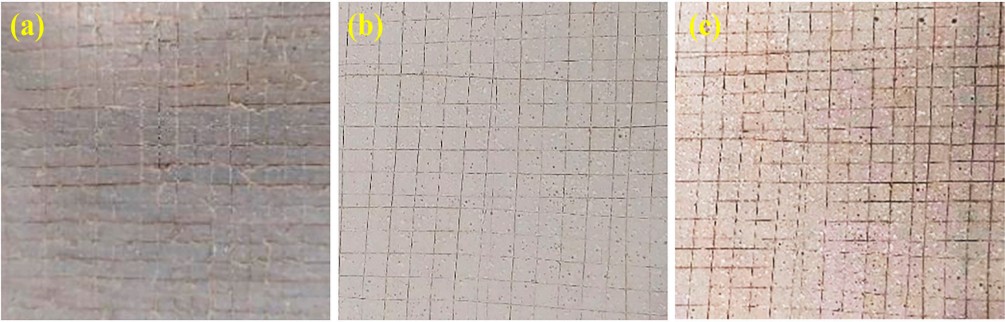

**Figure 7.** Bamboo scrimber surface coating adhesion test photo. Note: (**a**) 1%h-BN flame-retardant-treated bamboo scrimber; (**b**) 5% h-BN flame-retardant-treated bamboo scrimber; and (**c**) 10% h-BN flame-retardant-treated bamboo scrimber.

**Table 2.** Adhesion grade of the flame-retardant coating.

| Processing Method | Grade |
|---|---|
| 1% h-BN | 2 |
| 5% h-BN | 0 |
| 10%h-BN | 1 |

As illustrated in Figure 7, the 5% h-BN coating, with completely smooth cutting edges and without a single frame of peeling, achieved the highest adhesion level (Level 0), followed by the 10% h-BN coating, with a small amount of film peeling at the cutting intersections, with no more than 5% of the cut area affected, for which the adhesion Level 1 was determined. The 1% h-BN coating showed film peeling at the cutting edges, with more than 5% of the cut area affected, but, the adhesion of the 1% h-BN coatings with paint peeling off at the cutting edges affecting more than 5% of the cut area, but less than 15% of the cut area, was determined to belong to Level 2. The good adhesion of the coatings was due to the PVA component, which has good film-forming properties and some adhesion [27]. The h-BN concentration increased from 1% to 5%, and the adhesion of the coatings improved, indicating that the particulate structure of h-BN may increase the

bond with the film-forming substances, preventing the flow of molecules, which, in this study, led to an increase in the adhesion manifested in the better adhesion of the coatings. The coating's adhesion decreased at the h-BN concentration of 10% compared to the h-BN concentration of 5%, verifying that a high concentration of hexagonal boron nitride may lead to an aggregation phenomenon between particles, which reduces the fluidity and adhesion of the material.

## 4. Conclusions

Here, PVA and CS were adopted as film-forming substances, and h-BN flame-retardant coatings were prepared by adding different mass concentrations of h-BN. The flame-retardant coatings were formed on the bamboo scrimber surfaces, as confirmed by the FTIR spectra. The elemental distribution analysis indicated a more uniform dispersion of the 1% and 5% h-BN within the film-forming substances. The flame-retardant properties also tended to improve with the increase in h-BN concentration, which was consistent with the results of the thermal stability test. The cone calorimetry revealed that the 5% h-BN-treated specimen exhibited a 56% increase in their time to ignition, a 9.92% drop in their peak heat release rate, and a delayed time to reach their peak heat release rate. Although the peak smoke production rate increased by 17.78% owing to the thermal decomposition of the PVA ingredients, the presence of h-BN reduced the total smoke production rate by 32.35% during bamboo combustion, indicating an effective smoke-suppression. In the residual carbon structure after combustion, the charcoal layer with the 5% h-BN flame-retardant treatment was denser, which further proved that the flame-retardant and smoke-inhibition effect of the 5% h-BN coating was relatively better. The adhesion tests of the coatings at all three concentrations exhibited an excellent performance. Therefore, the flame-retardant and smoke-suppression properties of the bamboo scrimber treated with a 5% h-BN coating were better. This research on h-BN flame-retardant coatings provides a novel idea for future applications in the industrial field. The findings of the flame-retardant study in this experiment could be taken as an indication of the need for a more in-depth study of the flame-retardant and smoke-inhibition properties of h-BN coatings on bamboo scrimber to be conducted.

**Author Contributions:** Conceptualization, X.W.; methodology, X.W.; software, Y.Y.; validation, G.L.; formal analysis, Y.Y.; investigation, Y.Y.; resources, G.L.; data curation, G.L.; writing—original draft preparation, G.L.; writing—review and editing, X.W.; visualization, G.L.; supervision, S.Y. and W.Z.; project administration, W.Z.; funding acquisition, X.W. All authors have read and agreed to the published version of the manuscript.

**Funding:** This research was supported by the Zhejiang Provincial Project under Grant No. 2021F1065-3 and No. 2020TS09.

**Data Availability Statement:** All data can be found within the manuscript.

**Conflicts of Interest:** The authors declare no conflict of interest.

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
