# Peer review of "Flame-Retardant and Smoke-Suppression Properties of Bamboo Scrimber Coated with Hexagonal Boron Nitride"

_forests, doi:10.3390/f14102105_

Round 1

Reviewer 1 Report

In the title acronym is not recommended rewrite, also when mentioned first time in abstract or text write chemical name instead of acronym. 

In the abstract, Adhesion grade 0, what does it mean poor or good adhesion?

L 59 "some literature..." is poor information

L76 What material are we talking about the coating itself or coated wood?

In Table 1 "1788" what does that implicate?  

L97 h-BN solution was aqueous?

L105 how was samples prepared for FTIR (eg did it contain wood?)

L119: Nitrogen gas was used?

L132 How many grades and what is the basis for grading?

Sentence starting at L 132 a lot of peaks were given exact values but I can not see that from the spectrum, Make clear what is taken from literature and what was really observed in the spectrum.

L143. First part of sentence is just repetition of what already being said.

L151. The coated ones were compared with untreated but the latter was not shown in the figure, so how could that be understood? 

L 155 The agglomeration stressed by the author was not so evident, is it possible to mark it in the figure to make it more evident?

L 159 "poor" may be a strong word "less" is better as the shown area does not represent the total area of the sample.

L 192 I thought the increase from 5% to 10% was quite poor.

L.225 But increase higher than 5% was within experimental error.

L 233 1% and 10% seems to have increased compared to the control, rewrite!

Sentence starting at L234 is incomplete, adjust!

L236 for 1, 5 and 10% concentration?

L265 I doubt that the role PVA for smoke production was studied here as a separate test without presence of PVA, it is more based on earlier studies or knowledge.

Take away the last part of sentence as escape of personel was not studied and could be due also to other factors? or rewrite sentence with reference!

L. 282 middle of sentence should be "Figure 6"

L 289 How do we know it is related to h-BN?

L287 The results has already been published (ref 29)? make the information more clear!

L307 is a) in the Figure 6 untreated? and so on

L316. "The better adhesion..." was claimed but what is it compared with? also 1% h-BN had class 2 is it good or bad?

L339 "extremely obvious" is somewhat exaggerated.

English is quite OK some mistakes there are such as L 37 Take away the written "are". Please, check the manuscript as there are other small language errors!

Author Response

Reviewer 1:

  • In the title acronym is not recommended rewrite, also when mentioned first time in abstract or text write chemical name instead of acronym. 

Response: Thank reviewer for the advice. We have changed the acronym h-BN to the chemical name hexagonal boron nitride in line 2, line 14 and line 60.

  • In the abstract, Adhesion grade 0, what does it mean poor or good adhesion?

Response: Thank reviewer for the comment.The adhesion grade 0 means good adhesion. We have modified it in line 26 and completed the addition in line 147.

  • L 59 "some literature..." is poor information

Response: Thank reviewer for the remind. We have made changes in line 60 and added information about the relevant literature in line 63.

  • L76 What material are we talking about the coating itself or coated wood?

Response: Thank reviewer for the comment. We are talking about the coating itself in line 82.

  • In Table 1 "1788" what does that implicate?

Response: Thank reviewer for the comment. The 1788 represents the model of polyvinyl alcohol, and the PVA1788 represents the cold soluble PVA powder.

  • L97 h-BN solution was aqueous?

Response: Thank reviewer for the comment. The h-BN solution was aqueous in line 97. Sorry for the confusion caused to you due to the unclear expression.We have made changes in line 105.

  • L105 how was samples prepared for FTIR (eg did it contain wood?)

Response: Thank reviewer for the comment. The samples were pure coatings without bamboo scrimbers. The pure coatings were poured into clean glass petri dishes and dried in the oven.After that, the dried flakes were cut with scissors to a size of 5mm× 5mm.We have made changes in line 112.

  • L119: Nitrogen gas was used?

Response: Thank reviewer for the comment.The test was carried out under a dry nitrogen environment.We have completed the addition in line 128.

  • L132 How many grades and what is the basis for grading?

Response: Thank reviewer for the comment.The coating adhesion grade is divided into a total of 6 grades. The basis for grading is GB/T 4893.4-2013.This has already been explained in line 145.

  • Sentence starting at L 132 a lot of peaks were given exact values but I can not see that from the spectrum, Make clear what is taken from literature and what was really observed in the spectrum.

Response: Thank reviewer for the comment. These peaks in the infrared spectrum, while present in the spectrum, are really not very pronounced at 3261cm-1 and 2921cm-1, and stem more from references to the literature and then interpretations. The peaks at 1338cm-1 and 771cm-1 are noticeable, which is based more on actual observation.

  • First part of sentence is just repetition of what already being said.L143.

Response: Thank reviewer for the remind. We are very sorry for this error in the manuscript. We have made changes in line 154.

  • The coated ones were compared with untreated but the latter was not shown in the figure, so how could that be understood? 

Response: Thank reviewer for the comment. Your comment is so valuable and very helpful for revising and improving our paper. We have completed the addition in line 181.

  • L 159 "poor" may be a strong word "less" is better as the shown area does not represent the total area of the sample.

Response: Thank reviewer for the advice. We have changed in line 178.

  • L 192 I thought the increase from 5% to 10% was quite poor.

Response: Thank reviewer for the comment. Your comment is very helpful for revising and improving our paper. We have carefully considered your comment and carried out an in-depth analysis of bamboo scrimbers coated with 5% h-BN and 10% h-BN. After comprehensive consideration of factors such as pyrolysis rate and mass residual rate, we found that 10% thermal stability is still better than 5% thermal stability.

  • 225 But increase higher than 5% was within experimental error.

Response: Thank reviewer for the comment. We have made changes in line 240.

  • L 233 1% and 10% seems to have increased compared to the control, rewrite!

Response: Thank reviewer for the comment. We are very sorry for this error in the manuscript.We have made changes in line 248.

  • Sentence starting at L234 is incomplete, adjust!

Response: Thank reviewer for the comment. We are very sorry for such a mistake in the manuscript.We have made changes in line 245.

  • L236 for 1, 5 and 10% concentration?

Response: Thank reviewer for the comment. We are sorry for the confusion caused by our unclear description. The full statement would be “The occurrence time of exothermic peak of 1% h-BN,5% h-BN and 10% h-BN were prolonged by 11.32%, 41.51%, and 45.28%, respectively.” We have completed the addition in line 246.

  • L265 I doubt that the role PVA for smoke production was studied here as a separate test without presence of PVA, it is more based on earlier studies or knowledge.

Response: Thank reviewer for the comment. Your comment is very helpful for revising and improving our paper. We have thought deeply about your suggestion and will follow up with a separate study on the role of PVA in smoke production.

  • Take away the last part of sentence as escape of personel was not studied and could be due also to other factors? or rewrite sentence with reference!

 Response: Thank reviewer for the comment. We have thought deeply about your suggestion and made changes in line 280.

  • 282 middle of sentence should be "Figure 6"

Response: Thank reviewer for the comment. We apologise for our carelessness.We have changed "Figure 5" to "Figure 6" in line 309.

  • L 289 How do we know it is related to h-BN?

Response: Thank reviewer for the comment. Because among the components of the paint, only h-BN is white in colour, and h-BN has good heat-resistant properties[1].

  1. Liu, J.; Kutty, R.G.; Zheng, Q.; Eswariah, V.; Sreejith, S.; Liu, Z. Hexagonal Boron Nitride Nanosheets as High-Performance Binder-Free Fire-Resistant Wood Coatings. Small 2017, 13, 1602456, doi:10.1002/smll.201602456.
  • L287 The results has already been published (ref 29)? make the information more clear!

Response: Thank reviewer for the comment. The results of this study have not been published, and this article in the references describes one effect of pure h-BN nanosheets on the flame retardancy of wood[1].

  1. Liu, J.; Kutty, R.G.; Zheng, Q.; Eswariah, V.; Sreejith, S.; Liu, Z. Hexagonal Boron Nitride Nanosheets as High-Performance Binder-Free Fire-Resistant Wood Coatings. Small 2017, 13, 1602456, doi:10.1002/smll.201602456.

  • L307 is a) in the Figure 6 untreated? and so on

Response: Thank reviewer for the comment. We are sorry for the confusion caused by our unclear description.We have completed the addition in line 342.

  • "The better adhesion..." was claimed but what is it compared with? also 1% h-BN had class 2 is it good or bad?

Response: Thank reviewer for the comment.We apologise for our carelessness.We have made changes in line 350.The adhesion rating of 1% h-BN is good on its own, but inferior to that of 5% h-BN and 10%.

  • L339 "extremely obvious" is somewhat exaggerated.

Response: Thank reviewer for the comment.Your comment is very helpful for revising and improving our paper. We have changed in conclusions.

  • Comments on the Quality of English Language: English is quite OK some mistakes there are such as L 37 Take away the written "are". Please, check the manuscript as there are other small language errors!

Response: Thank reviewer for the comment.We apologise for our carelessness.We have made changes in line 40.We will check the manuscript carefully and correct small language errors in time.

Reviewer 2 Report

The manuscript presented a selection of fires retardant chemicals to improve the flame retardant properties of bamboo scrimber. The current topic is welcome in the context of sustainable industry.

Nevertheless, I have some questions and few recommendations for the authors:

Introduction

1.     At the beginning, please write in parenthesis with letters the name of all chemicals (where applicable), not only the formula, including hBN, because some readers could be not familiarized with fire retardant chemicals. Then you can use de symbols in the manuscript.

2.    Lines 76-80- Please reconsider the phrase or divide it in two sentences.

Method

3.    Line 85- please specify how many replicates did you used for each treatment/test?

4.    Lines 100-101- how many layers did you applied on surfaces and maybe it would be better if you express the amount as spreading rate in g/m2, as well.

5.    Line 129- The standard method must be titled.

Results

6.    Lines 151-165- You discussed about surface of bamboo scrimber, but in Figure 2 did not appear any image with uncoated bamboo scrimber to see the differences. Maybe, adding an image will be helpful for reader. Explain the figure 2, what means a1, a2, B, and the percents 16.8, 18, 17.4%, the correlation between images and concentration 1, 5, 10% of hBn?? Please refer figure in the text, otherwise it is difficult to follow the comments.

7.    Line 203-204- Figure 3- You did’ t note before the symbols URB, FRRB for treatments and they appeared in figure 3. Please return to the method and explain.

8.    Figure 6 -indicate on the figure the carbon residues. The text must be correlated with images and explained. What represents the percents from fig a, b, c, d4 because no reference is made in the text???  

9.    Was the combustion test done on a single sample/treatment? How statistically significant is the test?

10. Lines 312-315- Even that you followed a standard method maybe it would be good to explain the rating scale (0-5) for adhesion evaluation after cross-cut test. In how many places did you cut on each sample?  

11. Figure 7- the same magnification of the figures a,b c will be more relevant.

Conclusion

12. Lines 332-333- the first phrase is not completed.

13. Line 343-344- Revise de conclusion. Avoid exaggerated conclusions but describe the results according to the standard.

14. Maybe a recommendation for use in practice of fire retardants will be welcome for industry sector.

Author Response

Reviewer 2:

Introduction

  • At the beginning, please write in parenthesis with letters the name of all chemicals (where applicable), not only the formula, including hBN, because some readers could be not familiarized with fire retardant chemicals. Then you can use de symbols in the manuscript.

Response: Thank reviewer for the comment.Your comment is very helpful for revising and improving our paper. We have completed the addition in line 61.

  • Lines 76-80- Please reconsider the phrase or divide it in two sentences.

Response: Thank reviewer for the comment.We have made changes in line 80.

Method

  • Line 85- please specify how many replicates did you used for each treatment/test?

Response: Thank reviewer for the comment.We are so sorry we didn't make it clear how many times each test was repeated.The number of repetitions of the infrared test and the ring-scan energy spectrum test was 3. As the flame retardant test was pre-tested and found to be stable, it was carried out once during the formal test.

  • Lines 100-101- how many layers did you applied on surfaces and maybe it would be better if you express the amount as spreading rate in g/m2, as well.

Response: Thank reviewer for the comment.The method we use is the spray method, which controls the consistency of the paint by controlling its weight. We found the thickness of dry film reached to 0.3 mm ± 0.02 mm at room temperature.

  • Line 129- The standard method must be titled.

Response: Thank reviewer for the comment.We have changed in line 134.

Results

  • Lines 151-165- You discussed about surface of bamboo scrimber, but in Figure 2 did not appear any image with uncoated bamboo scrimber to see the differences. Maybe, adding an image will be helpful for reader. Explain the figure 2, what means a1, a2, B, and the percents 16.8, 18, 17.4%, the correlation between images and concentration 1, 5, 10% of hBn?? Please refer figure in the text, otherwise it is difficult to follow the comments.

Response: Thank reviewer for the comment.Your comments are very helpful in improving the quality of our manuscripts.We have added a SEM image of untreated  bamboo scrimber line 182.We have made additional changes to the content of the manuscript at line 167 in conjunction with the information in Figure 2 for a better understanding by the reader.

  • Line 203-204- Figure 3- You did’ t note before the symbols URB, FRRB for treatments and they appeared in figure 3. Please return to the method and explain.

Response: Thank reviewer for the comment.We have completed the addition in line 128.

  • Figure 6 -indicate on the figure the carbon residues. The text must be correlated with images and explained. What represents the percents from fig a, b, c, d4 because no reference is made in the text???  

Response: Thank reviewer for the comment.We apologise for any inconvenience caused by our inadequate presentation.The percents in the image represent the proportion of element B in the carbon residues.We have completed the addition in line 322.

  • Was the combustion test done on a single sample/treatment? How statistically significant is the test?

Response: Thank reviewer for the comment.A set of fire-retardant bamboo scrimber was indeed tested during the official test.We agree that more experiments will be necessary to understand the details of the experiment and to support the data. Because the experimental material bamboo scrimber itself has good homogeneity and denseness, and because we carried out precise control during the preparation of flame-retardant bamboo scrimber to ensure the consistency of the treatments, only one set of treatments was carried out for the formal experiments.This is consistent with the literature I've read on flame -retardant articles[1,2].We'll explore this in more depth later.

  1. Yang, F.; Hu, A.; Du, C.; Zhu, J.; Wang, Y.; Shao, Y.; Bao, Q.; Ran, Y. Preeminent Flame-Retardant and Smoke Suppression Properties of PCaAl-LDHs Nanostructures on Bamboo Scrimber. Molecules 2023, 28, 4542, doi:10.3390/molecules28114542.
  2. Du, C.G.; Song, J.G.; Chen, Y.X. The Effect of Applying Methods of Fire Retardant on Physical and Mechanical Properties of Bamboo Scrimber. AMR 2014, 1048, 465–468, doi:10.4028/www.scientific.net/AMR.1048.465.

  • Lines 312-315- Even that you followed a standard method maybe it would be good to explain the rating scale (0-5) for adhesion evaluation after cross-cut test. In how many places did you cut on each sample? 

Response: Thank reviewer for the comment.Your comment are vital to improving the quality of our manuscript.We have added the relevant details in line 337.For the experiment, a 3mm six-edged Baguette knife was used to make three cuts each in the horizontal and vertical directions of the coating, and the coating was observed to peel off.

  • Figure 7- the same magnification of the figures a,b c will be more relevant.

Response: Thank reviewer for the comment.We have changed in line 367.

  • Lines 332-333- the first phrase is not completed.

Response: Thank reviewer for the comment.We have changed the expression in line 350.

  • Line 343-344- Revise de conclusion. Avoid exaggerated conclusions but describe the results according to the standard.

Response: Thank reviewer for the comment.We have carefully revised the conclusions of the article in line 374 and line 379.

  • Maybe a recommendation for use in practice of fire retardants will be welcome for industry sector.

Response: Thank reviewer for the comment.We have completed the addition in line 389.

Reviewer 3 Report

The comments are attached

English must be improved significantly.

Author Response

Reviewer 3:

  • It is not clear to the readers what h-BN means. Please clarify in the title, the abstract and at the first mention in the text.

Response: Thank reviewer for the comment.We have completed the addition in line 14 and line 61.

  • The authors should improve the English language significantly. Some statements are difficult to understand.

Response: Thank reviewer for the comment.We will do our best to improve the manuscript and have made some changes to the problems with it. These changes will not affect the content or overall framework of the manuscript. We hope that the revised manuscript will be acceptable to you.

  • Please include the moisture content determination in methodology and results.

Response: Thank reviewer for the comment.Your comment is very helpful for revising and improving our paper. We have completed the addition in line 88.

  • CS and PVA do not melt (line 95) but perhaps dissolve. Please correct.

Response: Thank reviewer for the comment.We apologise for the inaccuracy of the expression due to our shallow knowledge and we have changed "melt" to "dissolved" in line 101.

  • The purpose of FR-IR analysis is not clear. Surely, each added ingredient will show or contribute to signals. The spectra were recorded at different concentrations of NB, but how the spectra changed is fully explained. For example, it is clear that the signal 3261 cm-1 is changed but this is not explained. The authors do not mention that PVA will also show some signals, not only BN and CS. The authors must detail how the samples were prepared and tested in FT-IR. The number of scans missing.

Response: Thank reviewer for the comment.We apologise for any confusion caused by our inadequate presentation.We have completed the addition in line 110 and line 113.

  • For TGA, it is not clear at all what atmosphere was used and what was the flow rate of the chosen gas. This should be detailed both in the methods section, in the results in relevant figure captions. The TGA of untreated wood must be included in the results. Please note there are no peaks on TG curves.

Response: Thank reviewer for the comment.Your suggestions are very helpful in improving the quality of our article.The TGA of untreated bamboo scrimber is in figure 3(b) .We have changed in line 127.

  • Cone calorimeter test should be more detailed. As the authors used wood material, which is very heterogeneous in nature, they should have carried out not a one-off cone test, but at least three and provide average values for TTI, HRR, THR, TSP etc. When detailing the size of the sample describe please do not use units for volume (mm3 ). Please correct throughout the text.

Response: Thank reviewer for the comment.We agree that more experiments will be necessary to understand the details of the experiment and to support the data.The experimental material of this experiment is bamboo scrimber, which is the use of bamboo bundles processed into bamboo filaments, after drying, glue dipping, and then drying the profiles made of high temperature and high pressure curing treatment. The material itself is better uniformity and denseness. In the flame retardant treatment of recombinant bamboo, it is also done with precise control, so the formal experiments are only a group of tests. We have changed the size of the description unit in line 91 and line 137.

  • Line 138, CS and BN cannot be named ‘monomers’. The authors did not carry out polymerisation. These are merely the ingredients of the coating.

Response: Thank reviewer for the comment.Your comment help us a lot to improve the quality of our article. Based on your comment, we have deleted “monomers”  in line 152.

  • SEM of untreated wood must be included for comparative purposes.

Response: Thank reviewer for the comment.We have completed the addition in line 182.

  • Line 175, TGA is not the method that confirms the synthesis of coatings. TGA simply measures the mass loss. It does not confirm the nature of coating. Please correct.

Response: Thank reviewer for the comment.We apologise for the inaccurate analysis. We have removed it in line 191.

  • Line 188, catalytic charcoal formation mentioned. Please explain what acts as the catalyst and what processes are involved.

Response: Thank reviewer for the comment.In the coating structure, h-BN has excellent unidirectional thermal conductivity [1], which can be dispersed into a layered structure in the film-forming material [2], prolonging the thermal conductivity path of the coating, effectively hindering the heat transfer, and acting as a catalyst for the carbon formation of the substrate material.

  1. Wang, J.; Ma, F.; Sun, M. Graphene, Hexagonal Boron Nitride, and Their Heterostructures: Properties and Applications. RSC Adv. 2017, 7, 16801–16822, doi:10.1039/C7RA00260B.
  2. Liu, J.; Kutty, R.G.; Zheng, Q.; Eswariah, V.; Sreejith, S.; Liu, Z. Hexagonal Boron Nitride Nanosheets as High-Performance Binder-Free Fire-Resistant Wood Coatings. Small 2017, 13, 1602456, doi:10.1002/smll.201602456.

  • Authors must explain why CS and PVA were selected. Do they interact as both are considered to be polyelectrolytes?

Response: Thank reviewer for the comment.We have already explained in the introductory section why CS and PVA have been chosen as the components of the film-forming substances. Firstly, Polyvinyl Alcohol and Chitosan are both green materials, Chitosan is inherently brittle due to its strong intramolecular and intermolecular hydrogen bonding and low crystallinity resulting in a CS film, while Polyvinyl Alcohol has good film-forming properties, chemical stability and strong adhesion.PVA and CS are often added and used as film-forming substances.[1,2]In this experiment, there was no chemical reaction between PVA and CS, and the dissolution of both was enhanced by heating in the presence of acetic acid solution, and physical interactions were formed between these components.

  1. Abraham, A.; Soloman, P.A.; Rejini, V.O. Preparation of Chitosan-Polyvinyl Alcohol Blends and Studies on Thermal and Mechanical Properties. Procedia Technology 2016, 24, 741–748, doi:10.1016/j.protcy.2016.05.206.
  2. Huang, M.; Fang, Y. Preparation, Characterization, and Properties of Chitosan-g-Poly(Vinyl Alcohol) Copolymer. Biopolymers 2006, 81, 160–166, doi:10.1002/bip.20383.
  • While discussing combustion parameters (section 3.4.1) the authors must report the average values with the relevant standard error.

Response: Thank reviewer for the comment.We agree that more experiments will be necessary to understand the details of the experiment and to support the data. Because the experimental material bamboo scrimber itself has good homogeneity and denseness, and because we carried out precise control during the preparation of flame-retardant bamboo scrimber to ensure the consistency of the treatments, only one set of treatments was carried out for the formal experiments.This is consistent with the literature I've read on flame -retardant articles[1,2].

  1. Yang, F.; Hu, A.; Du, C.; Zhu, J.; Wang, Y.; Shao, Y.; Bao, Q.; Ran, Y. Preeminent Flame-Retardant and Smoke Suppression Properties of PCaAl-LDHs Nanostructures on Bamboo Scrimber. Molecules 2023, 28, 4542, doi:10.3390/molecules28114542.
  2. Du, C.G.; Song, J.G.; Chen, Y.X. The Effect of Applying Methods of Fire Retardant on Physical and Mechanical Properties of Bamboo Scrimber. AMR 2014, 1048, 465–468, doi:10.4028/www.scientific.net/AMR.1048.465.

  • The cone data in Figure 5 is difficult to see. I suggest to make horizontal axis shorter in figures 5a and 5c, and to report SPR and TSP in the table.

Response: Thank reviewer for the comment.Your comments are very important to us.We've made changes to the chart's horizontal axis to allow the reader to make better observations in line 230.

  • Why authors to do report the yields of CO and CO2 recorded through cone?

Response: Thank reviewer for the comment.Your comment have been very helpful in improving the quality of our manuscripts.We have supplemented the manuscript with data on the yields of CO and CO2 in line 230.

  • It is incorrect to say that exothermic peaks are visible on HRR curves. Exo- or endothermicity is not evaluated through cone testing.

Response: Thank reviewer for the comment.We have changed in line 247 and line 245-255.

  • Lines 259-262, this statement does not make much sense as the authors compare the same parameter peak smoke release rate. Please revise.

Response: Thank reviewer for the comment.Your comment help us a lot to improve the quality of our article.After careful consideration, we have deleted the original first sentence, but considering that the second sentence is an introduction to the SPR values, we have retained it for better understanding by the reader. We have changed in line 269.

  • The authors did not use the correct methodology to determine the flame-retardant mechanism. They simply carried out SEM, this is not sufficient to make statements about the mechanism.

Response: Thank reviewer for the comment.We think your comment is very valuable, and we have thought deeply about it. Our initial intention is to prove the mechanism of flame retardation by observing the surface structure and internal morphology of the charcoal layer after combustion.When explaining the flame retardant mechanism of h-BN flame retardant coatings, we not only carried out visual inspection by electron microscopy, but also cited relevant literature to support the flame retardant mechanism of h-BN[1,2].In the following research, we will strengthen the study of the flame retardant mechanism through other test methods.

  1. Liu, J.; Kutty, R.G.; Zheng, Q.; Eswariah, V.; Sreejith, S.; Liu, Z. Hexagonal Boron Nitride Nanosheets as High-Performance Binder-Free Fire-Resistant Wood Coatings. Small 2017, 13, 1602456, doi:10.1002/smll.201602456.
  2. Wang, J.; Ma, F.; Sun, M. Graphene, Hexagonal Boron Nitride, and Their Heterostructures: Properties and Applications. RSC Adv. 2017, 7, 16801–16822, doi:10.1039/C7RA00260B.
  • Line 274, what ‘comprehensive smoke release rate’ means?

Response: Thank reviewer for the comment.We apologise for any inconvenience caused by unclear expressions.The phrase “Comprehensive smoke release rate” should be understood in conjunction with the following sentence “and total smoke production indicators”in the original manuscript.We have corrected line 288 to make it clearer to the reader.

  • Line 281, cone calorimeter should be used not conical calorimeter.

Response: Thank reviewer for the comment.We apologise for such a mistake.We have made a correction in line 308 and have checked the manuscript as a whole.

  • Conclusions must be re-written to include cone calorimetry data.

Response: Thank reviewer for the comment.We have revised the conclusion of the manuscript to include the cone calorimetry data in line 379.

Reviewer 4 Report

Review of manuscript forests-2624588

I am writing about the manuscript forests-2624588, “Flame-Retardant and Smoke Suppression Properties of Bamboo Scrimber Coated by h-BN”. It is an interesting study about using bornitrid at different concentration as coating substance for the scrimber.

I have small improvement suggestions regarding:

1.     In the Introduction section describe shortly what are scrimber and chitosan.

2.     The authors have introduced directly h-BN, also a short explanation of this abbreviation could help the reader better;

The manuscript needs a last check for proof reading.

Author Response

Reviewer 4:

  • In the Introduction section describe shortly what are scrimber and chitosan.

Response: Thank reviewer for the comment.We apologise for any inconvenience caused by the lack of sufficient information.We have completed the addition in line 68.

  • The authors have introduced directly h-BN, also a short explanation of this abbreviation could help the reader better.

Response: Thank reviewer for the comment.Your advice is crucial to improving the quality of our articles.We have completed the addition in line 61.

Round 2

Reviewer 2 Report

Please revise the sentence 1 from Conclusions!

Author Response

Response letter               Dear  Reviewers

We appreciate you very much for your positive and constructive comments on our manuscript.We have fully revised our manuscript and have addressed all of reviewer's comments.The detailed revisions are listed below and highlighted in the revised manuscript with green background.

Thanks for all the kind help.

Sincerely,

Xinzhou Wang *

College of Materials Science and Engineering,

Nanjing Forestry University,

Nanjing210037, PR China

Reviewer 2:

Introduction

  • Please revise the sentence 1 from Conclusions!

Response: Thank reviewer for the comment.Your comment is very helpful for revising and improving our manuscript. We have changed in line 389.

Reviewer 3 Report

The comments can be found in the file attached

Significant revision needed

Author Response

Response letter

Dear Reviewers

We appreciate you very much for your positive and constructive comments on our manuscript. We apologise for not having thoroughly revised the contents of the manuscript. We have re-corrected the manuscript. The detailed revisions are listed below and highlighted in the revised manuscript with green background.

Thanks for all the kind help.

Sincerely,

Xinzhou Wang *

College of Materials Science and Engineering,

Nanjing Forestry University,

Nanjing210037, PR China

Reviewer 3:

  • The signal 3261 cm-1 in FT-IR also relates to the presence of OH groups both in PVA and CS.

Response: Thank you for the advice. We apologize for not adding to this before. We have completed the addition in line 168.

  • The flow rate of nitrogen in TGA not given.

Response: Thank you for the advice. We apologize for not adding to this before. We have completed the addition in line 131.

  • It is my strong view that the authors did not carry out cone testing as per standard ISO5660-1. The samples were not conditioned prior to testing, the measurements were not repeated several times.

Response: Thank you for your serious comment. We have stated that the test was carried out in the spirit of ISO 5660-1 in line 141.

  • It is not scientific evidence to say in the response to the reviewer (not to the readers!) that ‘the experimental material bamboo scrimber itself has good homogeneity and denseness’. The authors should carry out cone calorimetry testing at least 3 times for each sample, and they will see how different the results would be. It is essential to carry out testing in triplicates (at least) and to provide the average results with the standard errors. The wood is a heterogeneous material, regardless of the tree species. The measurements will vary depending on the moisture content, which the authors failed to provide in the revised manuscript. Without these repeated tests the results lack scientific rigour required for publication in an OA journal.

Response: Thank you for your comments. First, we conducted a pre-test before the formal flame-retardant test and found that the flame retardancy of bamboo scrimber coated with h-BN was stable. It was probably due to the good homogeneity and compactness of the bamboo scrimber and the precise control during the preparation of the coating to ensure the consistency of treatments. Therefore, the test was carried out on only one sample per series, as is sometimes the case in flame-retardant tests [14,29]. At the same time, we have stated in the “2.3.4. Flame-retardant property test” on line 150-154 and “the conclusions” on line 407 that the flame-retardant data from this experiment is meant to be an indication, because in the future, we will conduct more experiments to understand the details of the experiments and to corroborate the data.

14 Du, C.G.; Song, J.G.; Chen, Y.X. The Effect of Applying Methods of Fire Retardant on Physical and Mechanical Properties of Bamboo Scrimber. AMR 2014, 1048, 465–468, doi:10.4028/www.scientific.net/AMR.1048.465.

29 Yang, F.; Hu, A.; Du, C.; Zhu, J.; Wang, Y.; Shao, Y.; Bao, Q.; Ran, Y. Preeminent Flame-Retardant and Smoke Suppression Properties of PCaAl-LDHs Nanostructures on Bamboo Scrimber. Molecules 2023, 28, 4542, doi:10.3390/molecules28114542.

  • The wood is also hydroscopic.

Response: Thank you for the advice. We have completed the addition in line 137.

  • flame retardant mechanisms cannot be deciphered by observing images of chars. Flame retardation occurs on the molecular levels, which cannot be captured by SEM. Establishing the mechanism of flame retardance is a complex task; a range of special hyphenated techniques is used such as Pyrolysis-GC/MS, TGA-GC-MS, NMR etc. The work of Camino, G., Schartel, B. and Morgan, A. would be very useful to the authors: https://www.sciencedirect.com/science/article/abs/pii/0141391088900730

Response: Thank you for the advice. The article you recommended is very helpful to me. In our further research, we will use more appropriate test methods to analyze the flame-retardant mechanisms of bamboo scrimber samples. So we have changed “3.4.2. Flame-retardant mechanism” to “3.4.2. Morphology of samples after flame retardancy tests ” in line 319 .
